# Cannibalism in the Brown Marmorated Stink Bug *Halyomorpha halys* (Stål)

**DOI:** 10.3390/insects11090643

**Published:** 2020-09-19

**Authors:** Giulia Papa, Ilaria Negri

**Affiliations:** Department of Sustainable Crop Production, Università Cattolica del Sacro Cuore Via Emilia Parmense 84, 29122 Piacenza, Italy; giulia.papa@unicatt.it

**Keywords:** cannibalism, *Halyomorpha halys*, Hemiptera, Pentatomidae, overwintering, phytophagous insect, diapause, aggregation

## Abstract

**Simple Summary:**

Overwintering populations of the crop pest *Halyomorpha halys* exhibit cannibalistic behaviour towards conspecifics. Depletion of metabolic reserves and desiccation occurring in winter can be overcome by intraspecific predation. This behaviour may be facilitated by the aggregation of individuals and the suppression of species-specific signals that prevent predation upon conspecifics.

**Abstract:**

The phytophagous brown marmorated stink bug *Halyomorpha halys* (Stål) is known to exhibit cannibalistic behaviour towards eggs. Here, we provide evidence of cannibalism among overwintering *H. halys* adults. Since diapausing individuals have high physiological demands for surviving long periods under stressful conditions, including the risk of depletion of metabolic reserves and desiccation, we assumed that nutritional and water requirements can be met by intraspecific predation. The role of aggregative behaviour in promoting cannibalism is also discussed. Given its evolutionary advantage, this trait should be maintained over generations and may be more widespread than previously considered in species that display aggregative behaviour during adverse seasons.

## 1. Introduction

The brown marmorated stink bug *Halyomorpha halys* (Stål, 1855) (Hemiptera: Pentatomidae) is native to East Asia, but has been spreading in Europe since 2004, first in Switzerland and then in the rest of Europe [1,2]. This species is a significant invasive agricultural pest that damages fruits trees (e.g., apples, peaches, and pears), vegetables, row crops, and ornamental plants [3,4]. *H. halys* feeds by penetrating a stylet bundle into plant tissues and injecting digestive enzymes to allow sucking of plant fluids, resulting in feeding injuries in the plant tissues, such as deformities, scars, discolouration, and pitting [4].

*H. halys* overwinter as adults and exhibit aggregative behaviour in thermally buffered microhabitats [5]. Aggregated individuals then enter facultative diapause for several months until specific environmental cues in spring trigger their re-emergence. For example, in northern Italy, adults exit diapause when the maximum temperature exceeds 14 °C and the photoperiod has 13 h of light [6]. Adults may become active through winter in response to warm temperatures and a poor nutritional state [7,8].

The metabolic strategies and physiological mechanisms used by *H. halys* to survive during winter diapause are not fully known [7]. According to previous studies, depletion of energy reserves and desiccation are the main causes of mortality in the overwintering generation, and adequate nutrition and sequestration of energy reserves (e.g., lipids, glycogen, and sugars) in individuals before entering diapause majorly influence diapause success [7,8,9].

Overwintering populations of herbivorous species displaying aggregative habits usually have few or no sources of food, other than dead conspecifics [10]. Conspecific necrophagy has been documented among overwintering adults, for example, in the boxelder bug *Leptocoris trivittatus* [10]. Necrophagy does not occur in *H. halys*, and freshly dead individuals can even repel conspecifics [11].

Here, we provide evidence that supports the hypothesis of cannibalism, i.e., feeding on live conspecifics, among overwintering adults of *H. halys*. Cannibalism is defined as intraspecific predation and has been observed in many animals, including protozoa, insects, birds, and mammals [12]. Cannibalism may play different roles in the ecology of a population, such as stabilising the host–plant/insect relationship, regulating population density, and providing a selective advantage by removing conspecifics that are infected with pathogens, which reduces the rates of parasitism in subsequent generations [13]. Cannibalistic behaviour may be influenced by both density-dependent and -independent factors; for example, developmental asynchrony with the host plant, sex or genetic relatedness of the participants, and limitations in the amount or quality of food and environmental factors, such as humidity and high temperature [13].

Although cannibalism has been observed in predatory insects, it has also been documented in many phytophagous species, including members of Lepidoptera, Coleoptera, and Hemiptera [12,13]. In phytophagous hemipterans, cannibalism has been reported in adults and nymphs which predate on eggs, e.g., *Parastrachia japonensis*, *Nezara viridula,* and *H. halys* [14,15,16,17], whereas cannibalism among adults has only been suggested but never documented [16].

## 2. Materials and Methods

During November 2017, about 400 *H. halys* adults in diapause at aggregation sites were collected from barns in the Province of Cremona (northern Italy, 45°08′17.3′′ N, 10°26′10.9′′ E). After removing dead/damaged individuals, 215 live *H. halys* adults were selected and placed into a single cage (ca. 40 cm high × 20 cm diameter) filled with cardboard panels. The cage was kept under outdoor conditions, following the protocol adopted by Costi et al. [6]. The individuals were maintained in diapause without any food or water for about 14 weeks. In mid-February 2018, diapause was terminated by exposure to 25 °C/L16:D8 to start laboratory colonies. Among individuals that had just exited diapause, we observed one living adult being cannibalised by a conspecific (Figure 1A; Appendix A). This observation was shortly followed by two other occurrences of this behaviour (Appendix A).

Dead specimens left in the overwintering cage were counted and the bodies inspected using a stereomicroscope to check for the presence of predation marks. Scanning electron microscopy (SEM; Zeiss Gemini SEM 500) was used to confirm the occurrence of predation marks. For SEM observations, naturally dehydrated specimens were gold-coated and mounted onto a carbon-coated stub.

In October 2018, a second population of *H. halys* adults (about 180) already triggered for diapause were collected from an overwintering site in the Province of Mantua (northern Italy, 45°00′14.9′′ N 11°04′38.5′′ E). After removing dead/unhealthy individuals, 147 live individuals were selected and maintained in diapause as previously described. In January 2019, dead and live specimens were counted. Dead specimens were inspected under a stereomicroscope for predation marks.

## 3. Results

In the first overwintering population, 90 out of 215 (about 42%) individuals were found dead; in the second population, 88 out of 147 (about 60%) were found dead. The stereomicroscope observations of the dead *H. halys* from the two populations identified 63 out of 90 and 23 out of 88 specimens, respectively, showing clear marks of cannibalism or feeding activity. The marks of cannibalism were in the form of perforation holes in the ventral part of the insect, while saliva residue on the stylet was considered a mark of feeding activity. Of the 63 specimens showing marks of cannibalism, 46 were putative prey, as they had perforation holes compatible with the predator stylet and were coated with coagulated substance compatible with haemolymph spilling (Figure 2B,C). Almost all predation marks were in the metathoracic ventral part at the level of the peritreme, median furrow, and ostiole of the scent gland (Figure 2A–C) or nearby joints of dorsal sclerites (Figure 1B). The remaining 17 specimens were putative predators since they showed presence of fluid exudates on the tip of the stylet. SEM observations confirmed the feeding holes in the fourth and fifth sternites (Figure 2E and Appendix A) and the presence of the salivary sheath on the stylet (Figure 3). The salivary sheath showed a typical bulbous morphology with sequentially stacked droplets (Figure 3). In the second population, SEM observations confirmed that all 23 specimens had predation holes.

Importantly, we also observed brownish faeces produced by allegedly starving individuals exiting diapause.

## 4. Discussion

In the present study, we provide evidence of previously undocumented behaviour in overwintering adults of *H. halys* that cannibalised other adults. Nutrient intake in diapausing adults has been previously documented in other insect species, such as the black blow fly *Phormia regina,* the mosquito *Culex pipiens*, and *L. trivittatus* [9,10]. Overwintering individuals undergo a costly physiological demand for surviving long periods of stressful conditions, including depletion of metabolic reserves and desiccation. The costs of diapause are commonly reflected in lower post-diapause survival and fitness. In *H. halys*, it has been estimated that more than 70% of adults do not survive winter and only 14% can survive until reproduction [6]. According to previous studies, the mortality of adults is due to a combination of dehydration and metabolic depletion of their energy reserves [9]. Our study supports the notion that nutritional and water requirements of overwintering adults can be met by intraspecific predation, and this could also explain why *H. halys* individuals that have just emerged from diapause have exceptionally high dispersal capacity, and can even survive for many days without food [18,19].

Successful feeding events are demonstrated by the formation of the salivary sheath at the tip of the stylet in some dead individuals (i.e., putative cannibals) [20]. As in other phytophagous hemipterans, feeding in *H. halys* is facilitated by the formation of a salivary sheath that encapsulates the stylet bundle [20,21]. As the stylet penetrates the tissue, *H. halys* secretes liquid droplets that solidify to form a solid hollow tube with typical bulbous morphology. The salivary sheath may stay attached to the feeding site, but in some cases, it may remain attached to the stylet when feeding ceases, being subsequently expelled during grooming [20,21]. In the present work, the overwintering populations were maintained in diapause without any food or water for about 14–16 weeks, and conspecifics were the sole source of food. At present, we do not know why these putative predators then died, and further work is needed to clarify this phenomenon.

In our opinion, cannibalism is also mediated by aggregative behaviour. Aggregation is widespread in the animal kingdom, especially in insects, and its ecological and evolutionary significance is based on the assumption that joining a group increases the survival and reproductive success of its members [22]. In insects, aggregation occurs via species-specific aggregation pheromones that induce group formation and attract and/or arrest conspecifics at the locality of release [22]. Aggregative behaviour in *H. halys* has been observed in both summer and overwintering generations [5]. The sex-specific aggregation pheromone is regularly released by males during the warmer months of the year to attract females [23,24]. Then, vibrational communications between the sexes facilitate mating [25]. In winter generations, sexually immature individuals aggregate in thermally buffered microhabitats that might also promote local moisture and water-conserving group effects [26,27]. Once diapause is initiated, *H. halys* males stop emitting aggregation pheromones and overwintering individuals cease to respond to experimental pheromone releases [28,29]. The unique volatiles released by diapausing adults consist of defensive compounds not specific to *H. halys* but commonly found in all stink bugs [29]. Since species-specific signals are known to prevent cannibalism in many animal taxa and may also help to avoid predation upon close relatives [30], we argue that the suppression of self-recognition mechanisms facilitates cannibalism in *H. halys*.

Our results suggest the novel conclusion that intraspecific predation also represents a strategy to cope with nutrient and water depletion. The ability to feed on conspecifics would also increase the probability of individuals surviving until host plants become available. We argue that, given its evolutionary advantage, this trait should be maintained over generations and might be more widespread in species displaying aggregative behaviour during adverse seasons than previously considered. During diapause, insects are not always totally inactive [31]. *H. halys* adults, for example, may become active through winter at temperatures above 9 °C [7,8]. In northern Italy, both 2017 and 2018 were characterised by exceptionally mild winters, with daily maximum temperatures in January below 13–14 °C.

Our experiment did not include continuous monitoring of the cages, hence it is possible that a proportion of those specimens that we classified as ‘cannibalised’ were subjected to necrophagy. In overwintering populations of species displaying aggregative behaviour, conspecific necrophagy may indeed provide resources during the winter months [10]. However, according to Chambers et al. [11], necrophagy is highly uncommon in *H. halys*. In their research, no evidence of necrophagy or survival advantage was recorded in overwintering individuals exposed to dead conspecifics during winter. In their experiments, the authors provided one-year-old dead *H. halys* to live individuals, but necrophagy was never observed [11]. In our opinion, the poor nutrient and water content of desiccated corpses cannot provide an appropriate food source, especially for a piercing-sucking species such as *H. halys*. Chambers et al. also reported that freshly killed individuals generated a repellent effect in live conspecifics [11]. The consumption of live conspecifics was observed in the present study. To the best of our knowledge, our study is the first to document cannibalism among adults in the phytophagous *H. halys*. Specific experiments are necessary to assess the extent of this behaviour, its sex bias, and the optimal microclimatic conditions that may trigger it.

We believe that shedding light on this undocumented behaviour might open the field to new and exciting research on the topic.

## 5. Conclusions

Cannibalism, i.e. intraspecific predation, is known to constitute an important component of the nutritional ecology of many species. In insects, cannibalism has been reported in non-carnivorous and strictly herbivorous species, where this behaviour provides a substantial nutritional benefit. Our results suggest that nutritional and water requirements of overwintering adults of the crop pest *Halyomorpha halys* can be met by intraspecific predation. Cannibals may therefore increase their probability to survive until host plants become available. This study provides a theoretical basis for a range of future studies to better understand the role of cannibalism in the biology and ecology of this invasive pest.

## Figures and Tables

**Figure 1 insects-11-00643-f001:**
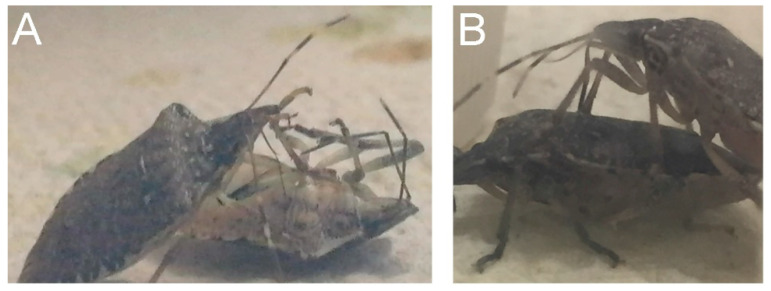
Prey and predators of *Halyomorpha halys*. (**A**) Frame of Appendix A showing cannibalism in the metathoracic ventral part of prey. (**B**) Picture showing predation in dorsal part of prey. Movement responses of prey prodded with a brush demonstrated that they were still alive (Appendix A).

**Figure 2 insects-11-00643-f002:**
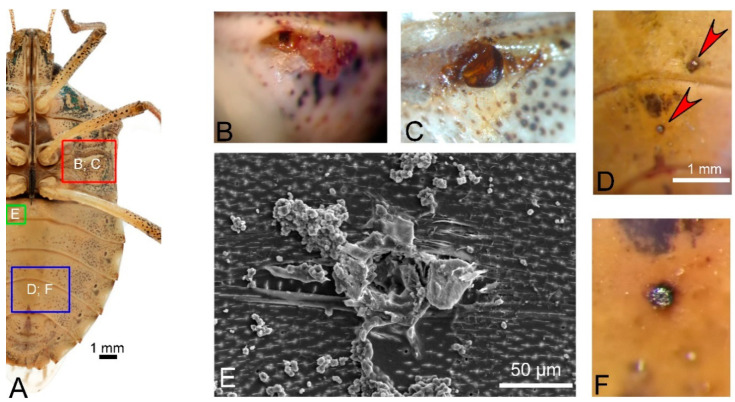
Stereomicroscope (**A**–**D**,**F**) and SEM (**E**) images showing marks of cannibalism in the ventral part of overwintering *Halyomorpha halys*. (**A**) The red square identifies the peritreme, median furrow, and ostiole of the scent gland area, and the green and blue squares highlight abdominal segments. Letters within squares refer to the position of the figures. (**B**) Holes with tissue and haemolymph residues. (**C**) Haemolymph clot. (**D**) The red arrows show the holes in the abdomen. (**E**) SEM image of the hole in the abdomen. This picture represents the same individual in Appendix A. (**F**) Close-up image of the holes in D.

**Figure 3 insects-11-00643-f003:**
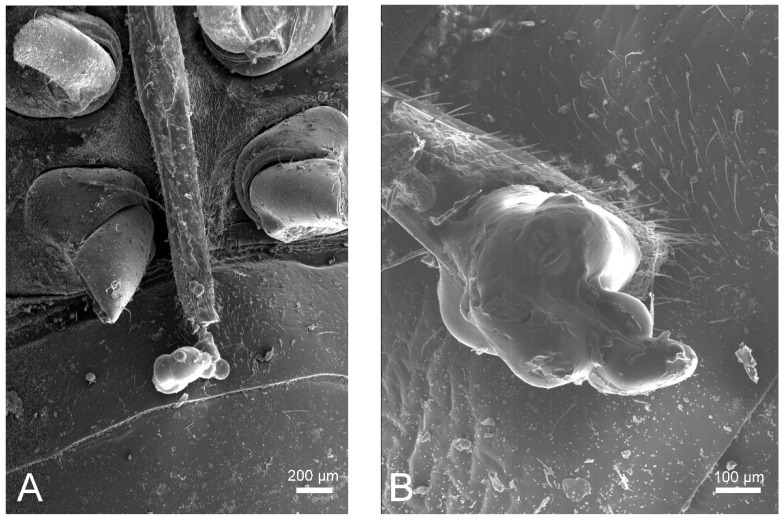
SEM images of the stylet of overwintering *Halyomorpha halys* associated with cannibalism. (**A**) Stylet with salivary sheath. (**B**) Close-up image of a salivary sheath showing the typical bulbous morphology.

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
