# Peer review of "Cannibalism in the Brown Marmorated Stink Bug Halyomorpha halys (Stål)"

_insects, 2020, doi:10.3390/insects11090643_

Round 1
Reviewer 1 Report
The authors report for the first time the observation of cannibalistic behavior in Halyomorpha halys, based in studies performed in two populations collected in the same geographical area. This finding is novel and interesting, and it could set ground for future studies on the conditioning aspects and ecological relevance of this behavior.
I recommend this study for publication in Insects, upon clarification of minor comments:
1.- I would like to have access to specific information about how many barns were sourced to collect the population from 2017, and which specific location was used in 2018. Could it be that we are talking about different cohorts of the same population? I think this is relevant at the time of concluding how spread the cannibalistic behavior can be in genetically related populations/groups.
2.- I find that the authors could improve the clarity of their reported results on the number of individuals with cannibalistic marks:
2.1.- Line 82 reads that the first population "90 out of 215 (about 42%) individuals were found dead" and line 84 "63 out of 125 [...] showing clear marks of cannibalism" Wouldn't it be 63 out of 90?
2.2.- Line 84, regarding the second population: "23 out of 88 specimens, respectively, showing clear marks of cannibalism" but line 96 reads "In the second population, all the 88 dead specimens showed holes of predation". Now, I understand this difference could be due to a more sensible read out with SEM: the first set of data was obtained with stereoscope examination, whereas the second was obtained upon SEM observation. If this is the case, it should be clearly stated in the results section.
2.3.- If SEM is a more resolutive observation technique, and that lead to an increase on the number of individuals with cannibalistic marks among the second population, why didn't those numbers increased on the first population as well after examination with SEM? Do they remain as a total of 63 out of 90? This contradictory observations should be clarified, or reported in more clearly.
3.- I am pleased that the authors addressed the possibility of necrophilic events in those populations. I would suggest that they emphasize the differences between this study and the one from Chambers and colleagues: those two populations of H. halys are very separated geographically, and adapted to very a different environments and weather. On top of that, if I understood the experimental designed properly, the animals were kept under very different light conditions between these two studies, being in "outdoor conditions" here, and in complete darkness in Virginia. All these differences can affect the behavior of the populations, and I think this should be addressed in the discussion. I would not discard (and neither did the authors) the presence of necrophilic behaviors. Maybe in the future the authors could consider videotaping the experiments, to have access to real time data, instead of analyzing only the end results.
Reviewer 2 Report
This paper provides information suggesting the occurrence of cannibalistic behavior of overwintering adult BMSB on conspecific adults. Data is given that suggests feeding wounds on dead BMSB and corresponding signs of hemolymph exudate on feeding sheaths of some other adults. A video is provided that shows one such attack. This data will be of interest to the BMSB research community. However, the authors overstate their feeding wound data as proof of cannibalism – conclusive proof requires additional data showing that the exudate on stylet sheaths is from BMSB, or additional visual confirmations of actual feeding, or both. Nevertheless, the study is interesting and may be published following some revision.
Comments by line number below point out a number of grammatical and other edits for proper English.
Line
Simple Summary
9 exhibits should be exhibit (not plural); and pest crop is reversed – should be crop pest
10 reserves should be plural: reserves
Abstract
15&17 semicolons used incorrectly - replace with commas
Main text
39 insert “these” before “authors”
43 reverse the order of have & usually
48&66&78&79 replace alive with live (live is used instead of alive when applied as an adjective that precedes a noun – i.e., live bug vs. bug that was alive)
69 replace On with In
78 replace not healthy with unhealthy
80 replace “through stereoscope” with “using a stereomicroscope”. A stereoscope is any device, not necessarily a microscope, by which two images of the same object taken at slightly different angles are viewed to produce a stereo view.
83 see above
85&6 Specific preyed-upon individuals and the presumed predatory individuals that fed upon them are two separate categories need to be reported as separate numbers. Line 85 is misleading by referring to both groups together as having marks of cannibalism. As written this sentence implies that all were prey.
90 insert “the” between of and peritreme
92-93 Were any specimens examined of BMSB that had recently fed upon plant hosts to compare the appearance of fluids at the tips of stylets? How can authors be reasonably certain that the stylets with fluids were fresh enough to have come only from cannibalism during overwintering? Discussion on this would be helpful.
95-96 What have SEM observations confirmed concerning the stylet sheaths? This part of the sentence is incomplete. It should say “confirmed presence of fluid exudates on stylet tip” or something like this.
100 in figure caption, replace preys with prey (prey is both singular & plural in English)
104 in figure caption, replace stereoscope with stereomicroscope.
116 The authors overstate their results here – there was only a single example of visual proof a BMSB feeding upon another (S1) claimed to be from a live prey. This was not apparent in the video – how did authors know the prey BMSB was still alive when fed on? If there is corresponding proof this should be mentioned. The other puncture data provided are highly suggestive of additional conspecific feeding events, but conclusive proof would require further visual observations, or molecular confirmation of BMSB fluids on stylet sheaths. These suggestions should be clearly presented. Furthermore, the authors make a point of distinguishing between cannibalism of live BMSB and feeding on dead BMSB. It is not apparent from their Methods and discussion that these two states could be disguished in their observations. Perhaps more discussion of this can be presented to avoid this problem All in all these observations are interesting and form the basis for an excellent followup study.
118 replace have with has
124 insert “these” or “the cited” before authors
125 replace “the hypothesis” with “our hypothesis”
131 replace “As all” with “As in other”
146 Replace Then with “Subsequently,”
171&172 replace alive with live
Reviewer 3 Report
Cannibalism in the Brown Marmorated Stink Bug Halyomorpha halys (Stål).
This work provides evidence for cannibalism among adult Stink Bug Halyomorpha halys adults while overwintering. The general topic is of interest and appropriate for a short communication. However, several aspects of the methodology should be further clarified and justified, and interpretation should be restricted accordingly. For example, under what conditions were the insects kept? Were these equivalent to the conditions they experience in the field, or could cannibalism have been an artifact of laboratory conditions? Could some of the cannibalism have occurred following diapause termination, or were the live individuals immediately removed? Could some of the feeding events occurred after the individuals were already dead (i.e., necrophagy)? Although you claim that this possibility was overruled in another study, conditions and insect strains might have been different. Also, it is not clear why should the bugs avoid already dead individuals, which could have provided them with a source of energy. While the suggestion that cannibalism might be common in aggregating insects, no test of this is given (e.g., comparison of cannibalism rates under different densities, comparison of aggregative vs. non-aggregating insects etc.). In addition, it should be acknowledged that while cannibalism is indeed be very beneficial for the cannibalizing individual, it is obviously not so for the cannibalized one. Hence, high risk of cannibalism may also potentially select against aggregation, possibly making this trait combination less common.
Specific comments:
Abstract
Line 19: Change to "the risk of depletion…"
Line 14: Change to "are known to exhibit…"
Line 16: Change "of" to "under"
Line 17: You do not really test this hypothesis. You show that cannibalism (probably) occurs, but whether this provides the nutritional and water requirements is still an assumption.
Line 16-17: Change to "is native to"
Introduction
Line 29: Grapefruit can be considered as another a fruit tree
Line 36: Change "and" to "combined with"
Line 39: Change to "according to previous studies"
Line 40: Change "cause" to "causes"
Line 41: Change to: "energy reserves (e.g., lipids, glycogen and sugars), before the entry…"
Line 42: Add comma after "diapause", change "in" to "on"
Lines 43-44: Why is this the case? Is it because there are normally no sources of food in the aggregation sites, or because population density is too high?
Lines 45-46: Can you know for sure that it never occurs? What could be the reason?
Line 48: Cannibalism is a phenomenon, not really a hypothesis.
Line 54-57: Shortly explain how these factors were shown to affect cannibalism rates.
Line 58: Change to "predatory insects"
Line 59: Change "like" to "including members of"
Line 60: Could be useful to point out that no eggs are available during diapause (if true)
Materials and Methods
Line 65: Could have been useful to inspect individuals already dead when collected for feeding marks.
Line 68: More information is needed - at what kind of container/conditions/densities, were insects kept? Were these equivalent to conditions in the field?
Line 70: Could some of the feeding events happened after they exited diapause?
Line 71: You mention one event where cannibalism was directly observed but present two pictures (in figure 1). When did the second event occur?
Line 76: What is the assumption that they were "already triggered for diapause" based on?
Lines 70 and 79: Inspecting still live individuals could have also been useful. I guess this was not possible as they were used to start a colony?
Results
This paragraph is very difficult to follow. I suggest to present the data in a table. Also, I would present the two types of evidence (holes and residue on stylet) separately, as they imply very different things.
Were there any individuals with both prey and predator marks?
Did you record the sex of the individuals?
Line 94: Why is this assumed to be produced by starved individuals and what is the importance of this?
Discussion
Line 116: I would change "proof" to "evidence"
Line 117: "nutrient intake" in what form?
Line 124: Change "authors" to "previous studies"
Line 125: Change "following our observations" to "our study supports the notion"
Lines 125-129: I'd break into two sentences.
Line 130: "also" in addition to what?
Line 135: No need for a new paragraph here.
Line 139-154: This paragraph raises interesting topics, but it is largely unrelated to the results of this study and hence should be shortened and restricted. You do not test in any way the connection between aggregation and cannibalistic behavior or the recognition mechanisms.
Line 156: While the ability to feed on conspecifics increases survival prospects, the risk of cannibalism also reduces survival prospects of individuals in the population. This might select against aggregation.
Line 164-172: I think that necrophagy cannot be entirely overruled based on the results of a previous study.
Line 175: Do you have information on the sex of individuals?
